# Multiscale Very High Resolution Topographic Models in Alpine Ecology: Pros and Cons of Airborne LiDAR and Drone-Based Stereo-Photogrammetry Technologies

**Annie S. Guillaume** [1], **Kevin Leempoel** [1,2], **Estelle Rochat** [1], **Aude Rogivue** [3], **Michel Kasser** [4], **Felix Gugerli** [3], **Christian Parisod** [5] and **Stéphane Joost** [1,*]

1    Laboratory of Geographic Information Systems (LASIG), École Polytechnique Fédérale de Lausanne, 1015 Lausanne, Switzerland; annie.guillaume@epfl.ch (A.S.G.); k.leempoel@kew.org (K.L.); estelle.rochat@alumni.epfl.ch (E.R.)
2    Royal Botanic Gardens, Kew Richmond TW9 3AE, UK
3    WSL Swiss Federal Research Institute, 8903 Birmensdorf, Switzerland; aude.rogivue@agroscope.admin.ch (A.R.); felix.gugerli@wsl.ch (F.G.)
4    Haute Ecole d'Ingénierie et de Gestion du Canton de Vaud, 1400 Yverdon-les-Bains, Switzerland; michel.kasser@heig-vd.ch
5    Institute of Plant Sciences, University of Bern, 3013 Bern, Switzerland; christian.parisod@ips.unibe.ch
*    Correspondence: stephane.joost@epfl.ch; Tel.: +41-21-693-57-82

**Abstract:** The vulnerability of alpine environments to climate change presses an urgent need to accurately model and understand these ecosystems. Popularity in the use of digital elevation models (DEMs) to derive proxy environmental variables has increased over the past decade, particularly as DEMs are relatively cheaply acquired at very high resolutions (VHR; <1 m spatial resolution). Here, we implement a multiscale framework and compare DEM-derived variables produced by Light Detection and Ranging (LiDAR) and stereo-photogrammetry (PHOTO) methods, with the aim of assessing their relevance and utility in species distribution modelling (SDM). Using a case study on the arctic-alpine plant, *Arabis alpina*, in two valleys in the western Swiss Alps, we show that both LiDAR and PHOTO technologies can be relevant for producing DEM-derived variables for use in SDMs. We demonstrate that PHOTO DEMs, up to a spatial resolution of at least 1 m, rivalled the accuracy of LiDAR DEMs, largely owing to the customizability of PHOTO DEMs to the study sites compared to commercially available LiDAR DEMs. We obtained DEMs at spatial resolutions of 6.25 cm–8 m for PHOTO and 50 cm–32 m for LiDAR, where we determined that the optimal spatial resolutions of DEM-derived variables in SDM were between 1 and 32 m, depending on the variable and site characteristics. We found that the reduced extent of PHOTO DEMs altered the calculations of all derived variables, which had particular consequences on their relevance at the site with heterogenous terrain. However, for the homogenous site, SDMs based on PHOTO-derived variables generally had higher predictive powers than those derived from LiDAR at matching resolutions. From our results, we recommend carefully considering the required DEM extent to produce relevant derived variables. We also advocate implementing a multiscale framework to appropriately assess the ecological relevance of derived variables, where we caution against the use of VHR-DEMs finer than 50 cm in such studies.

**Keywords:** alpine ecology; *Arabis alpina*; digital elevation models (DEMs); light detection and ranging (LiDAR); multiscale; photogrammetry; spatial scale; species distribution models (SDM); terrain attributes; very high resolution

## 1. Introduction

Alpine environments are among the most sensitive ecosystems to climate change and associated extreme weather fluctuations [1]. Increases in mean annual air temperature coupled with changes in precipitation patterns have been associated with glacial

retreats, permafrost degradation, and increases in sedimentation and erosion, leading to an upward migration of vegetation belts and increased interspecies competition over the past 100 years [2–6]. As such, there is a growing interest in alpine conservation research to understand how plants are likely to respond to novel pressures, and with this, guide appropriate and effective management strategies.

Modelling species distributions in alpine environments requires a move from the traditional methods that rely on environmental and climatic data interpolated from regional-scale sensors, such as local weather stations [7,8]. New evidence highlights the importance of incorporating fine-scale topographic information for modelling microclimatic conditions [9,10], as microclimates are thought to largely determine habitat availability, the number of species a region can support, and the ability of species to either stay or go under a changing climate [7–9,11].

Recently, ecologists have been turning towards remote sensed digital elevation models (DEMs) to obtain accurate topographical attributes that proxy for environmental variables [12,13]. Originally adopted in the fields of geology and hydrology to describe terrain steepness and orientation, the primary terrain attributes of slope, aspect, and curvature derived from DEMs can be used to calculate more complex secondary terrain attributes. These secondary derived variables have been developed to accurately model ecologically-relevant environmental factors, such as soil depth, nutrient status, solar radiation, terrain ruggedness, humidity, and soil wetness [7,10,12,14–16], for example, which have been successfully used to model species distributions [7,17,18], for studying responses to environmental change [8,9,11], and for evaluating capacities for local adaptation [19].

Accurate modelling of microhabitat conditions requires very high resolution (VHR) DEMs of 1 m or finer, which have recently become more financially and logistically obtainable. There are two main capture systems to produce VHR-DEMs: airborne Light Detection and Ranging (LiDAR) and drone-based stereo-photogrammetry (PHOTO). While LiDAR is traditionally the method used to acquire DEMs, particularly as it produces highly accurate models under most weather, light, and vegetation conditions [20–22], popularity is increasing for PHOTO as a practical and cheaper solution to LiDAR [23–25], particularly in alpine terrain above the tree line where the models are directly of the surface [26], despite PHOTO's limitations regarding sensitivity to terrain complexity, vegetation, lighting, and weather conditions [27,28]. Technical differences between the technologies result in an important trade-off between the maximum extent captured during a flight and the final spatial resolution [23,29]: LiDAR's use of an active laser system restricts it to airborne platforms, resulting in larger extents at resolutions typically around 50cm, while the camera equipment of the passive PHOTO method allows its use on drones, restricting extent while improving spatial resolution by up to 10 times that of LiDAR (i.e., finer than 5 cm).

Acquiring VHR-DEMs raises important questions relating to accuracy and scale, such as: is such level of detail necessary for ecological studies? And: what resolution is precise enough to adequately represent topography and microclimates? Indeed, DEM resolutions that are too fine may hold an excess of detail, while those that are too coarse may over-generalize and lose important features [30,31]. The optimal spatial resolution depends on the derived variable: predictions of soil depth, for example, were found to be most accurate when using DEMs at 2 m resolution [32], while models of topsoil pH were optimized when using DEMs at 1 m resolution [33]. Similar patterns are found for climatic variables, where one study found optimal resolutions to be between 0.5 and 4 m [15]. Unsurprisingly, species distribution models (SDMs) have been shown to depend on the spatial resolution of the input variables [18,34]. Yet, a general lack of guidance and formal discussion around the topic of spatial scale has resulted in the arbitrary selection of resolutions in many ecology studies [35,36], with consequences for ecological modelling [37,38].

Here, we implement a multiscale framework and compare DEM-derived variables produced by LiDAR and PHOTO technologies on the same two alpine regions, with the aims of: (i) assessing the predictive powers of variables derived from each technology, (ii) illustrating the importance of mindfully selecting spatial scale, and (iii) highlighting

key differences between the two technologies. We implement SDMs with data from a case study on *Arabis alpina* in the western Swiss Alps to evaluate the ecological relevance of DEM-derived variables at multiple scales from both technologies. We conclude by providing a comparison of LiDAR- and PHOTO-based DEMs for use in ecological research, and we outline key differences between the technologies to help researchers decide on the appropriate DEM-acquisition technology to use in their research.

## 2. Materials and Methods

We obtained VHR (<1 m resolution) DEMs for two alpine sites located in the western Swiss Alps, in the context of a landscape genomics project (GENESCALE) [25,39,40]. We were interested in testing the adequacy of the cheaper and logistically simpler PHOTO method using a drone compared to state-obtained LiDAR using airplanes, with the motivation for cost-effective repeatability of the project in other areas. This presented a good opportunity to compare DEMs produced by LiDAR and PHOTO technologies, as well as their associated derived variables, particularly as a lack of vegetation and man-made structures at the study sites meant that the digital surface model (DSM) obtained by PHOTO made it equivalent to a terrain model [26]. To avoid ambiguity, we use the term *spatial resolution* to describe the pixel size of DEMs and derived variables, while we use the term *extent* to describe the total area sampled or analyzed [36].

### 2.1. Study Sites

Both La Para (Para) and Col des Martinets (Martinets) target study sites cover 0.5 km$^2$ and are located in alpine valleys located above the tree line in the western Swiss Alps (Figure 1). Para is located in a narrow valley and is characterized by bumpy and steep slopes; Martinets is located at a slightly higher elevation in a wider valley and is characterized by flatter terrain with a cliff across the middle of the site (Table 1). These sites were selected for this case study as they show a diverse range of alpine habitat types (e.g., rocks, grassy-meadows, low shrub, cliff-faces) with high topographic complexity [25]. A total of 181 and 123 geo-referenced points ($\pm$3 cm) were recorded across Para and Martinets, respectively, using a Leica DGPS (Table 1). Of these, 24 and 13 points, respectively, were measured specifically as recognizable ground control points for producing PHOTO DEMs using Global Navigation Satellite Systems in Real-Time Kinematic mode (GNSS RTK), while 146 and 100 points were geo-referenced locations of the alpine perennial plant, *A. alpina* (Brassicaceae). The remaining geo-referenced points (n = 11 and 10, respectively) correspond to the location of environmental loggers, whose data were not used in the present study. Location of plant and logger points were selected across the sites to represent contrasting microhabitats within the context of the landscape genomics project.

### 2.2. Digital Elevation Models

2.2.1. LiDAR Acquisition

Raw LiDAR point clouds were obtained from a laser survey carried out by the Direction of Land registry and of Geoinformation (DCG), of the Swiss state of Vaud in June 2015, and distributed by ASIT Vaud [41]. The point cloud was acquired using a LiDAR Optech ALTM Gemini at a wavelength of 1064 nm and a flight altitude of approximately 650 m above the terrain. The laser used a scanning angle of ~20° on both sides of the vertical and resulted in a point density of between 8 to 12 points m$^{-2}$.

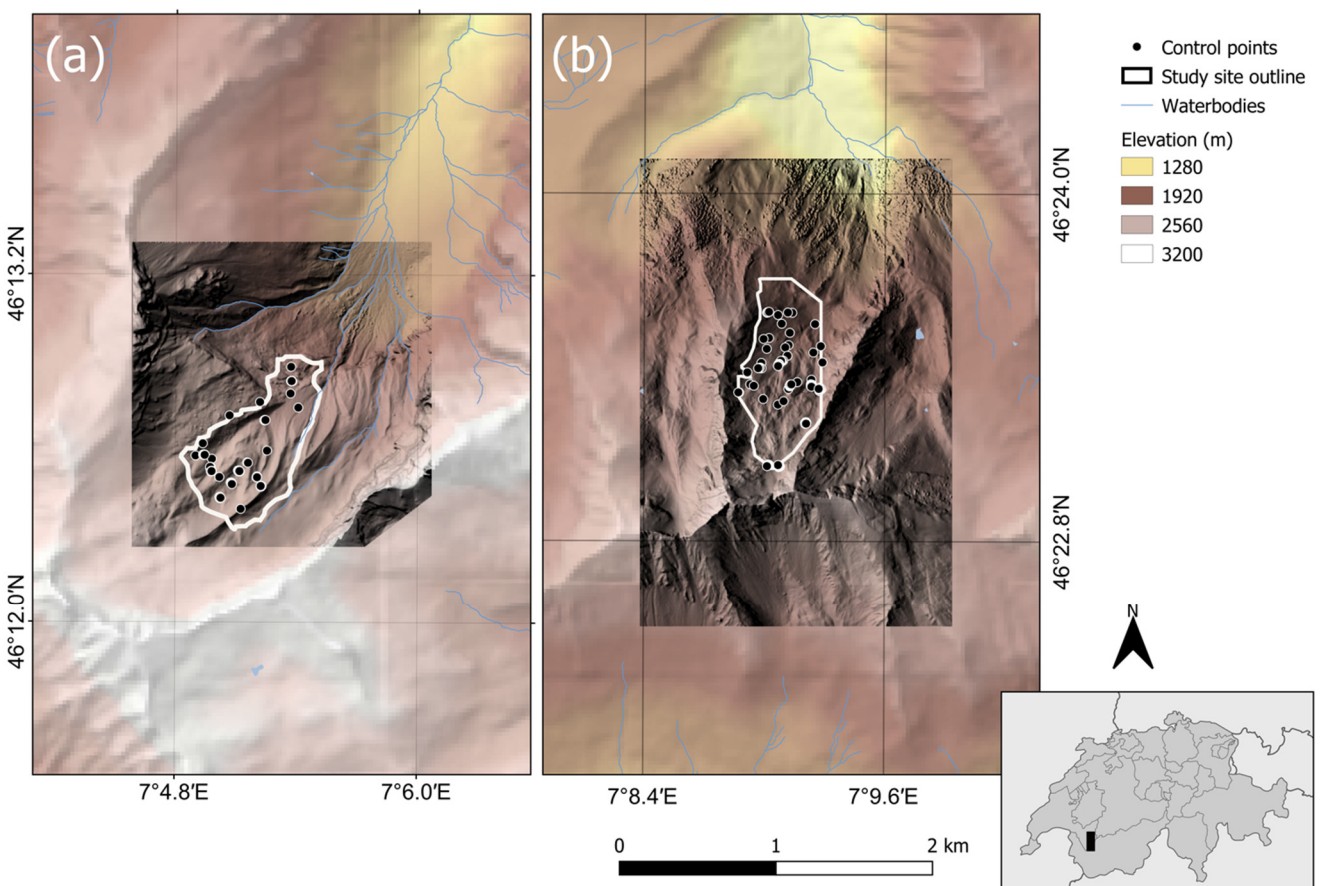

**Figure 1.** Location of (**a**) Martinets and (**b**) Para study sites in the western Swiss Alps (location indicated in black rectangle on the Swiss map insert). The site perimeter is outlined in white and geo-referenced points are indicated as black dots. The Light Detection and Ranging (LiDAR) digital elevation models (DEMs) are superimposed on an elevation map of the surrounding area and are in darker colors.

**Table 1.** Location and characterization of the two study sites, Martinets and Para, situated in valleys in the western Swiss Alps, including the areas of the target study sites and DEM extents, as well as the number of geo-referenced points used in the analyses.

|  | **Martinets** | **Para** |
|---|---|---|
| Coordinates | 46°12′37″N; 7°5′12″E | 46°23′23″N; 7°9′6″E |
| Elevation range | 1928–2368 m asl | 1826–2320 m asl |
| Orientation | NE | NNE |
| Slope (mean ± sd) | 0.45 ± 0.17 rad | 0.50 ± 0.16 rad |
| Eastness (mean ± sd) | 0.34 ± 0.6 rad | 0.44 ± 0.5 rad |
| Northness (mean ± sd) | 0.48 ± 0.6 rad | 0.64 ± 0.4 rad |
| VRM [1] (mean ± sd) | $4.7 \times 10^{-3} \pm 9.6 \times 10^{-3}$ | $4.5 \times 10^{-3} \pm 7.3 \times 10^{-3}$ |
| Area of LiDAR DEM | 3.7 km$^2$ | 6.0 km$^2$ |
| Area of PHOTO DEM | 0.7 km$^2$ | 0.7 km$^2$ |
| Area of target site | 0.5 km$^2$ | 0.5 km$^2$ |
| Ground control points | 13 | 24 |
| Plant occurrence points | 100 | 146 |
| Logger points | 10 | 11 |
| Assessment points [2] | 110 | 157 |

[1] VRM = Vector Ruggedness Measure (no unit). [2] Sum of plant occurrence points and logger points, used for assessing DEM accuracy.

### 2.2.2. Photogrammetry Acquisition

PHOTO data were collected in August 2014 using a SenseFly eBee fixed-wing UAV [42], which is capable of autonomous flight in wind speeds up to 50 km h$^{-1}$ (rapid wind and weather changes are common at altitude in this region; actual windspeed of approximately 35 km h$^{-1}$ was recorded during flights). A flight plan was developed to account for a 130 m flight-height above the terrain aiming for a ground pixel resolution of about 4 cm, as well as a 75% longitudinal and 60% lateral overlap of images to allow for good stereoscopy. Parameters were slightly modified when flown over certain terrain types, particularly cliff areas, to ensure a homogeneous ground pixel size throughout the flight. Images were analyzed and orthorectified using Pix4D software (v1.1.45, 2014). Images were initially processed at double image size to maximize extracted features and improve accuracies, after which, additional 3D points were computed based on the original image size to maximize point cloud densification, and the minimum number of matches per 3D point was set to three. The DSM was produced at a resolution of 5cm per pixel using inverse distance weighting, where noise and errors in calculated points were filtered using the median elevation of neighboring points, and small bumps were removed using the medium surface smoothing filter. This was exported as a *.las* file. All PHOTO DEM acquisition and processing was performed at HEIG-VD, Switzerland.

### 2.2.3. DEM Processing

We used CloudCompare (v 2.10.2, 2020) to process the point cloud *.las* files from both technologies. The *Rasterize* tool was used to produce the VHR-DEMs from the point clouds using average cell height, where missing data were filled using linear interpolation with values from the nearest neighboring cells. The finest resolution DEM from the LiDAR point cloud was 0.5 m (the finest resolution available from ASIT Vaud [41]), while the finest resolution DEM for PHOTO was 6.25 cm (the finest resolution available from the drone flight that was a factor of 0.5m to allow for comparison with LiDAR). The DEMs were modified in R (v3.6.0, 2019) to add empty pixels along the right and bottom borders, such that the raster extents were divisible by $2^n$ for multiscale decomposition up to *n* times. All DEMs were geo-referenced in the Swiss reference system (MN95: CH1903+/LV95). See Table 1 for the areas of the finest-resolution DEMs obtained for LiDAR and PHOTO at both sites, as well as the areas of the target study sites.

### 2.2.4. DEM Multiscale Decomposition

A multiscale framework was used to investigate the effect of spatial scale on the accuracy of DEMs obtained by LiDAR and PHOTO technologies, as well as to assess the optimal spatial scales of derived variables. We generalized the DEMs to multiple scales using the Gaussian Pyramid algorithm in MATLAB with the *impyramid* function (MathWorks: MATLAB R2019a, 2019) [43–45], which reduces image size by half, each step. The PHOTO DEM was generalized from 6.25 cm to 12.5 cm, 25 cm, 50 cm, 1 m, 2 m, 4 m, and 8 m, and the LiDAR DEM was generalized from 50 cm to 1 m, 2 m, 4 m, 8 m, 16 m, and 32 m, where the maximum resolution of 32 m was selected, as it covers the commonly used resolution of 25 m in ecological studies (e.g., [10]). The differences in the ranges of DEM spatial resolutions produced for LiDAR and PHOTO are due to the trade-off between the finest obtainable pixel size and the extent of the area captured. The generalized DEM rasters were then manually cropped using QGIS (v3.4, 2019) to remove incorrectly calculated border pixels, due to edge effects during generalization.

### 2.2.5. DEM Accuracy Assessment

The accuracy of each DEM (LiDAR at 7 resolutions; PHOTO at 8 resolutions) was evaluated as the vertical error ($\Delta h_i$) of the elevation obtained from the DEM compared to the accurately measured assessment points ($\Delta h_i$ = Assessment point – DEM point; Para: n = 157, Martinets: n = 110), following the DEM accuracy assessments as recommended by [46]. All calculations were performed in R. The distribution of $\Delta h_i$ was first visualized

using boxplots and quantile–quantile (Q-Q) plots to check for deviations from a normal distribution. Vertical accuracy was assessed using standard accuracy measures (assuming a normal distribution of errors), including mean, standard deviation (sd), and root mean square error (RMSE), as well as assessed with robust measures of accuracy, including minimum, maximum, median, and the normalized median absolute deviation (NMAD)—an estimate for the standard deviation that is more resilient to outliers. Outliers were defined as points with a $|\Delta h_i|$ greater than three times the RMSE [47].

### 2.3. DEM-Derived Variables

#### 2.3.1. Derived Variable Computation

A total of 23 variables relating to terrain morphometry, hydrology, and solar radiation were derived from each DEM using SAGA GIS (v7.5.0, 2019). Eastness and Northness were calculated as the sine and cosine of Aspect in R. Descriptions and parameters used in calculations of all the derived variables are provided in Table S1 in the Supplementary Materials.

#### 2.3.2. Derived Variable Correlations

Correlations between each pair of derived variables generated at 0.5m resolution were calculated to select independent variables following rules adapted from [15]. Spearman's rank correlations were calculated based on a subset of 15,000 random points across each study site. Using a correlation threshold of $|r_s| \geq 0.8$ [48], we reduced the number of variables to eight (Table 2), prioritizing primary terrain attributes (slope, aspect, curvature), followed by variables that were deemed to be more ecologically meaningful to high-altitude alpine plants [49]. For each DEM, we reassessed pairwise correlations for the eight independent variables to investigate how spatial resolution, technology, and site characteristics alter collinearity. Additionally, we produced scatterplots to directly compare the eight independent LiDAR and PHOTO DEM-derived variables at the common spatial resolutions of 0.5 m, 1 m, 2 m, 4 m, and 8 m.

**Table 2.** Description of independent DEM-derived variables computed at each resolution for LiDAR and PHOTO. See Table S1 in the supplementary material for the parameters used in calculations.

| | Variable | Abbv. | Description | Units | Ref. |
|---|---|---|---|---|---|
| **Primary attributes** | Elevation | Elev. | DEM elevation, interpolated from LiDAR or PHOTO, generalized to multiple resolutions using B-spline wavelet transforms. | m | [42,43] |
| | Slope | Slope | *Morphometry*. Local morphometric terrain parameters; proxies for water flow, snow movements, erosion, solar radiation, etc. Eastness and Northness represent the sine and cosine of Aspect (Orientation), respectively. Curvature is used to understand erosion and runoff processes. | radians | |
| | Eastness | East | | radians | [50] |
| | Northness | North | | radians | |
| | Plan curvature | Hcu | | *1/m* | |
| **Secondary attributes** | Vector ruggedness measure | VRM | *Morphometry*. Quantifies rugosity with less correlation to slope, indicating a combined variability in slope and aspect. | No unit | [51] |
| | SAGA wetness index | SWI | *Hydrology*. Modified version of Topographic Wetness Index (TWI), which is a calculation of the slope and a modified catchment area (MCa). It predicts a more accurate soil moisture for cells situated on the valley floor (when compared to the TWI) | MCa/ Slope | [52,53] |
| | Sky view factor | SVF | *Lighting*. Ratio of the radiation received by a planar surface to the radiation emitted by the entire hemispheric environment | No unit | [14,54,55] |
| | Total Solar radiation in June | Ti06 | *Lighting*. Sum of direct and diffuse insolation in summer (calculated for 1 to 30 June 2015). | kWh/m$^2$ | [12,14] |

#### 2.3.3. Derived Variables in Species Distribution Models

To investigate the ecological relevance of DEM-derived variables from LiDAR and PHOTO at various spatial resolutions, we performed SDMs at each site following the methods of [56]. As we had presence-only data of *A. alpina* across the sites, we used the machine learning method of MaxEnt [57] through the R package, *maxnet* [58], to estimate the probability distribution n of the plants across the sites based on incomplete species presence-only data and environmental predictor variables [59,60].

For each model, we used Elevation and the eight independent variables from Table 2, varying only the DEM-acquisition technology and spatial resolution, where all rasters were limited to the extent of the study site, as indicated in the maps of Figure 2. To determine the technology–resolution combination that can best discriminate plant location from random background points for each variable at each site, we performed Student *t*-tests in R with the *t.test* function, comparing the plant presence data (Para: n = 146, Martinets: n = 100; green dots in Figure 2) to 10,000 random background points at each site. We considered a technology–resolution combination to be significant if the *t*-test value was <0.01 after applying a Bonferroni correction for multiple comparisons. For each variable at each site, we retained the technology–resolution combination that produced the largest *T*-value and used these in a "combination" model for each site (see Table S2 for summary of *t*-tests results).

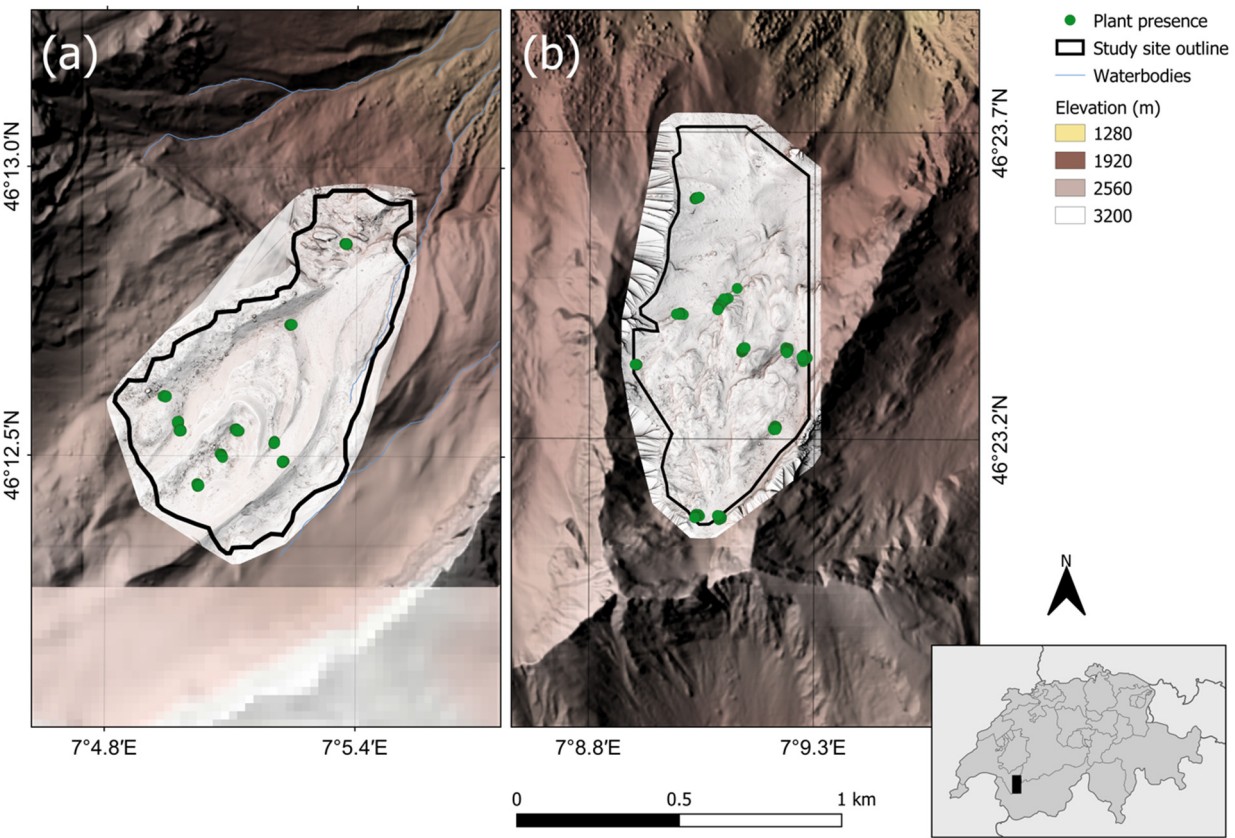

**Figure 2.** Location of *Arabis alpina* individuals (green dots) across (**a**) Martinets (n = 100 locations) and (**b**) Para (n = 146 locations) sites, where target site perimeters are outlined in black. The sky view factor (SVF) variable derived from PHOTO is shown in light colors and superimposed on hill shaded DEMs.

As species' responses to environmental factors tend to be complex [59], MaxEnt allows for non-linear transformations, termed feature classes (FC) of predictor variables, which are regulated for overfitting with regularization multipliers (RM) [60]. To optimize these two parameters for our SDMs, we assessed the performance of MaxEnt models produced for each site using the combination models specific for each site, where we varied the FC transformations (linear; linear-product; linear-quadratic; linear-product-quadratic) and RM values (1, 2, 5, and 10). We ran 20 models each with a different random subset, using 75% of the data to train the models and 25% to test them, and projected the models using the cloglog scaled output. Two metrics were used to evaluate the performance of the MaxEnt models [61]: the mean of the commonly used Area Under the Receiver Operating Curve (AUC) [62], based on the test data (AUCtest), which was complemented with the mean sample-size corrected Akaike Information Criterion (AICc) [63]. From this we determined

that the optimal parameters for the SDM at both sites was the linear-product-quadratic FC coupled with a RM of 1 (LPQ1; Table S3). Finally, we performed MaxEnt SDM for *A. alpina* at both study sites separately, using the optimized parameters to transform the predictor variables. We performed one model for each technology–resolution combination, resulting in a total of 15 models in addition to the combination model, per site. Though at the coarser resolutions there were instances of multiple plant occurrences per grid, we retained all presence locations to maintain consistent sample sizes in the models with different spatial resolutions [61].

## 3. Results

### 3.1. DEM Accuracy Assessment

DEM accuracy ($\Delta h_i$), measured as DEM vertical error when compared to assessment points, decreased at coarser spatial resolutions, particularly from 4 m onwards (Figure 3; Table S4). However, accuracy remained relatively constant from 6.25 cm through to 50 cm.

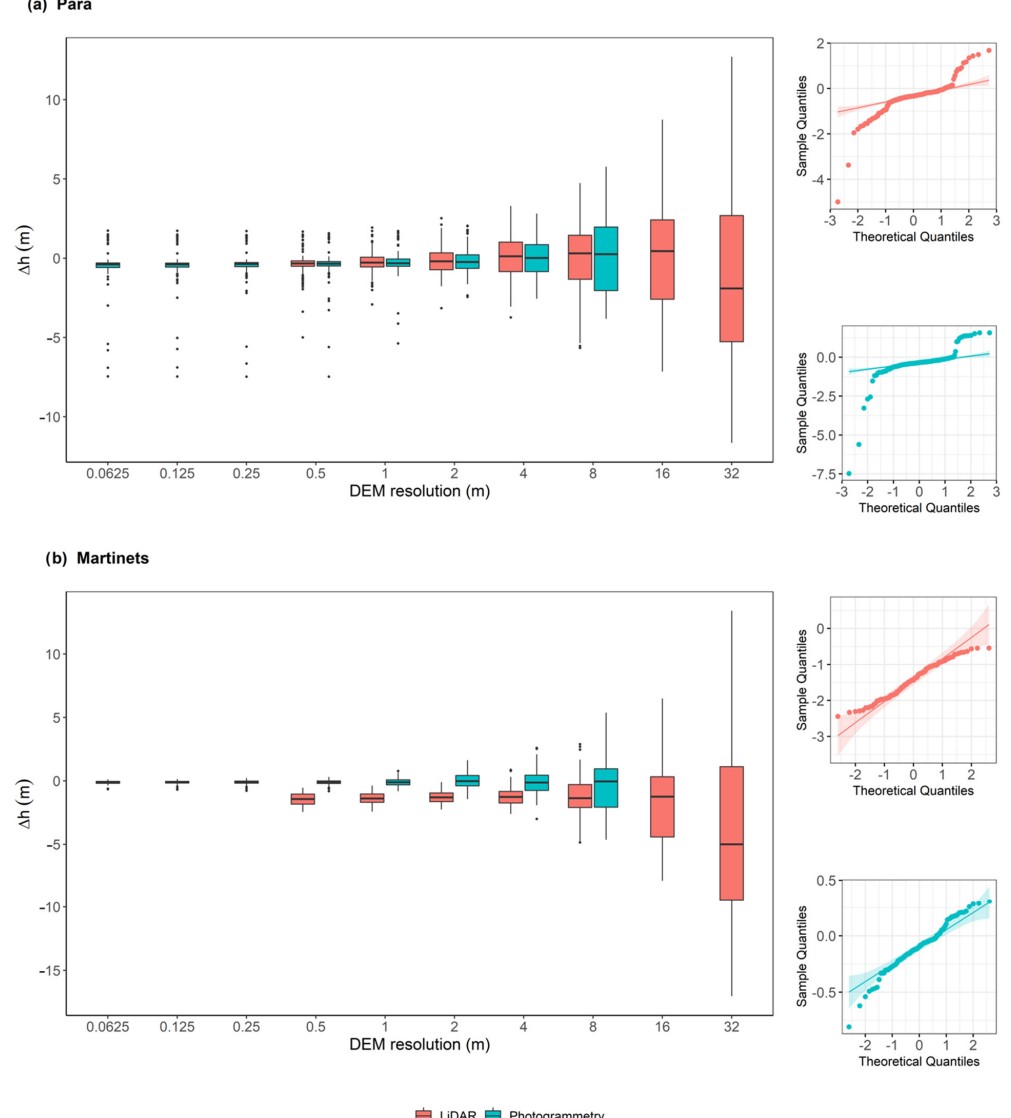

**Figure 3.** The vertical error (Δh; in meters) of DEMs acquired from LiDAR (orange) or PHOTO (blue) technologies across all resolutions for (**a**) Para (n = 157) and (**b**) Martinets (n = 110). Boxplots are accompanied by examples of Q–Q plots of LiDAR and PHOTO DEM errors at 0.5m resolution. The complete set of Q–Q plots for both sites can be found in Figure S1.

Visual inspection of the vertical error distribution using Q–Q plots (Figure 3 for Q–Q plots of 0.5 m DEMs; Figure S1 for all Q–Q plots) showed strong deviations from a normal distribution, where heavy tails at the finer resolutions indicate the presence of extreme values, such as outliers. These extreme values diminished at coarser resolutions. Skews in the data were stronger at Para than at Martinets, with four outliers detected at Para compared with two at Martinets. We assessed accuracy using robust measures that are more resistant to outliers, though we report both standard and robust measures in Table S4.

The accuracy of PHOTO DEMs was equal to, if not better than, the accuracy of the LiDAR DEMs for spatial resolutions up to 1 m at Martinets and 4 m at Para. At coarser resolutions for both sites, PHOTO DEMs became more varied and less accurate than LiDAR. At Para, LiDAR and PHOTO DEMs showed similar magnitudes of error, where PHOTO had slightly more outliers than LiDAR at matching resolutions (Figure 3a, Table S4a). Additionally, PHOTO DEMs at Para had approximately the same variation as LiDAR DEMs at matching resolutions, as determined by NMAD, the robust estimate for standard deviation (Table S4, Figure S2a). At Martinets, PHOTO DEMs were more accurate than the LiDAR DEMs when compared to the assessment points (Figure 3b, Table S4b), where the 0.5 m PHOTO DEM over-estimated elevation by a median of 10 cm, while the 0.5 m LiDAR DEM over-estimated elevation by a median of 1.4 m. Additionally, PHOTO DEMs at Martinets showed less variation in error than did LiDAR DEMs up to resolutions of 1 m (Table S4, Figure S2b), after which vertical errors in PHOTO DEM were slightly more variable than LiDAR.

### 3.2. DEM-Derived Variables

#### 3.2.1. Derived Variable Correlations

Eight independent DEM-derived variables were retained from an initial 23, based on a Spearman rank correlation threshold of $|r_s| \geq 0.8$ for the variables at 0.5 m resolution (Table 2). When correlations were reassessed at coarser resolutions, most correlations remained consistent and below the $|r_s| = 0.8$ threshold (Table S5).

The data showed strong correlations between the DEMs of LiDAR and PHOTO at all resolutions ($r_s = 1$), but this was not observed for the derived variables between the technologies (Figure S3). Indeed, there was a high degree of scatter, indicating inconsistencies between variables derived from the LiDAR versus PHOTO DEMs, particularly at finer resolutions. This was especially notable for HCu and VRM at both sites, where variables were poorly correlated at 0.5 m resolutions ($r_s < 0.2$ and $<0.45$, respectively), but were more aligned at 8 m resolutions ($r_s > 0.8$ and $>0.89$, respectively). For other variables, including East, North, and Ti06, LiDAR and PHOTO produced relatively congruent variables at both sites, with $r_s > 0.75$ at the 0.5m resolution and $r_s > 0.9$ at the 8 m resolution. In general, variables derived from LiDAR and PHOTO DEMs were slightly more consistent for Para than Martinets.

#### 3.2.2. Predictive Power of Derived Variables in Species Distribution Models

The optimal resolution–technology combination that distinguished plant presence from random background points was dependent on the site characteristics (Table S2). At Para, 120 of the 135 variable–resolution combinations (9 variables each at 15 resolutions) were able to significantly distinguish plant locations from random background points at a significance level of $p \leq 0.05$, where more derived variables were from LiDAR than from PHOTO DEMs. As such, all derived variables used in the "combination" MaxEnt model for Para were LiDAR-derived. The optimal spatial resolution depended on the variable: Elevation and Slope at 32 m; VRM at 16m; Hcu, SVF and SWI at 8 m; Ti06 at 4 m; and East and North at 1 m. At Martinets, 101 of the 135 variable–resolution combinations were able to distinguish plant location from random points, with weaker significances than at Para. Interestingly, 59 variables significantly able to distinguish plants from background were PHOTO-derived, while 42 were LiDAR-derived. Despite this, all variables that could most significantly distinguish plant location were LiDAR-derived, with the exception of

SVF, where most variables were optimized at the coarser resolutions: Elevation, East, Hcu, Slope, and Ti06 at 32 m; SWI at 16m; SVF at 4 m; and North and VRM at 1 m. These optimal variables for each site, as listed above and highlighted in grey in Table S2, were then used to determine that the optimal MaxEnt model parameters to use are a FC of linear-product-quadratic coupled with a RM of 1 (LPQ1; Table S3).

The ability of each MaxEnt model to predict the distribution of *A. alpina* at each site was assessed using AUCtest and AICc (Figure 4). At Para, all models appeared to be well suited to predicting plant location, with mean AUCtest values >0.8 (Figure 4a). Model performance at Para improved with variables at 6.25 cm to 12.5 cm, after which SDM performance was approximately equal with variables at 12.5 cm to 4 m resolutions (Figure 4a,c). The highest predictive power was the model produced with 16m resolution variables, followed by the Para combination model. MaxEnt model performance at Para was slightly improved when variables were derived from LiDAR rather than from PHOTO. SDMs at Martinets were marginally less accurate than at Para, with mean AUCtest values >0.75. While model performance improved as input variable resolution coarsened, particularly from 6.25 cm to 1 m, there was little difference in model performance from 2 m to 32 m (Figure 4b,d). Highest model performance was obtained with the combination model, followed by 4 m PHOTO, and 32 m LiDAR. With the exception of 1 m resolution, SDMs based on PHOTO-derived variables had higher predictive power than those based on LiDAR at Martinets.

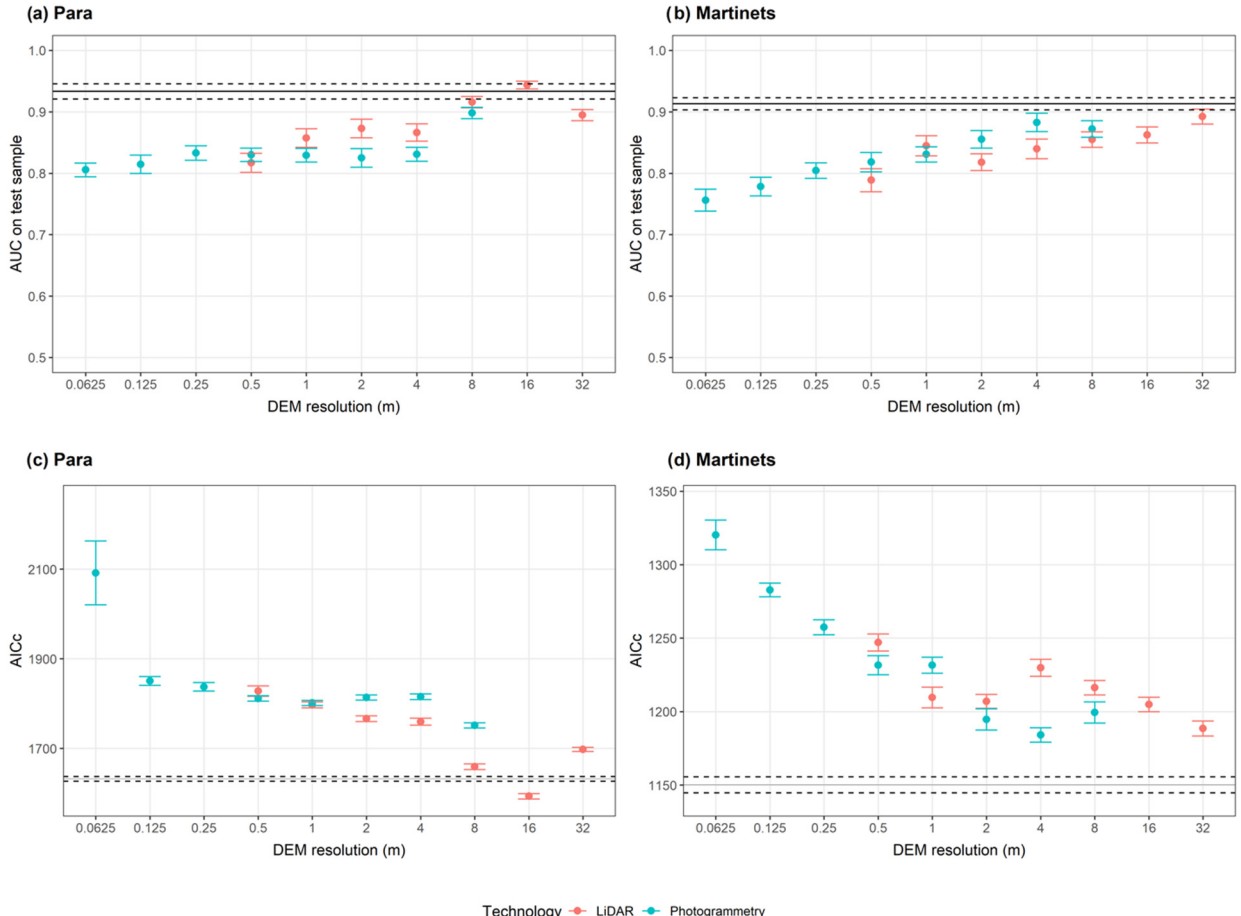

**Figure 4.** MaxEnt SDM performances evaluated using AUCtest and AICc at Para ((**a**,**c**), respectively) and Martinets ((**b**,**d**), respectively) as a function of variable resolution (*x*-axes) and DEM-acquisition technology (LiDAR in orange, PHOTO in blue). Each point represents the evaluation criteria mean ± sd, based on the results of 20 model iterations for the input variables. The horizontal black lines represent the mean ± sd of the "combination" model calculated for each site using the variable resolutions that best differentiated plant occurrence points from random background points.

## 4. Discussion

Using a multiscale framework to model plant distributions across two alpine study sites, we systematically evaluated the ecological relevance of DEMs, and derived variables acquired from both LiDAR and PHOTO technologies.

Here, the accuracy of PHOTO DEMs rivalled that of LiDAR DEMs, putting the current paradigm of LiDAR being the most accurate DEM-acquisition method into question. Furthermore, we show that the optimal spatial resolutions for DEM-derived variables in alpine plant distribution modelling is between 1 and 32 m, depending on the variable and the characteristics of the study site. This reiterates concerns regarding the use of the finest obtainable resolutions (<0.5 m) to represent micro-climatic conditions experienced by sessile organisms, further supporting suggestions that such high resolutions are simply introducing noise to ecological models [18]. Here, we discuss the influence of spatial scale and DEM-acquisition technologies on the relevance of derived variables in SDM at the two study sites, after which we provide an overview of the technologies to assist in selecting the most appropriate method for producing data to use in alpine ecology studies.

### 4.1. Spatial Scale

Through a multiscale framework, we show that DEM accuracy is stable up to 2 m resolution, then reduces at coarser resolutions. This corroborates previous studies [31,33,44] that show how DEM generalization smooths over topography, removing information about microtopography and noise from outliers that are present at the finer resolutions.

Despite higher accuracies of the VHR-DEMs, SDMs using variables at coarser resolutions between 1 and 32 m best predicted the distribution of *A. alpina* across the study sites. Indeed, DEM-derived variables for use in ecological modelling need to be at spatial resolutions that accurately represent climatic variables, such as air temperature, humidity, and soil moisture, where it has been shown in a similar Alpine region that climatic variables are best represented by derived variables between 1 and 4 m [15]. It should be noted that despite relatively lower performance, the SDMs built with VHR input variables remained useful models of plant distribution. Additionally, it has been shown that the number of location points and their clustering across a site can alter the performance of SDMs [61], which may influence optimal spatial resolution for SDMs.

The optimal spatial resolution depended strongly on the derived variable and characteristics of the study site [15,61]. Indeed, there was some improvement in SDM when each input variable was used at its specific optimal resolution in the "combination" model. Additionally, SDMs for more flat and homogeneous topography variables, as at Martinets, were optimized with coarser resolution, while SDMs for more complex heterogenous areas, such as Para, were optimized with a combination of coarse- and fine-resolution variables, supporting conclusions from previous studies [62].

We demonstrate that the relationships between the eight retained independent variables changed with spatial resolution and characteristics of the study sites. For example, the relationship between SVF and East was stronger at Para, with its NNE orientation, than at Martinets, with its NE orientation, where the correlation between North with SVF weakened as the spatial resolution coarsened.

### 4.2. LiDAR Versus Photogrammetry

At matching resolutions, PHOTO DEMs rivalled the accuracy of LiDAR DEMs, where PHOTO DEMs up to at least 1 m resolution were in fact more accurate than the LiDAR DEMs. There are two main reasons why we might have seen this result. First, the use of the logger and plant points to measure DEM accuracy favors the PHOTO DEMs, as these points were collected at the same time that the PHOTO DEM was produced, while the LiDAR DEM was collected as part of a state-wide elevation campaign and may have differences in calibration. Second, the two DEMs were collected at different times of the year: while PHOTO was collected in August, LiDAR was collected in June of the following year, at a time when snow was likely to be covering parts of these high-altitude sites. We found

that at Martinets in particular, the LiDAR DEM over-estimated elevation by approximately 1.4 m when compared to the assessment points at 0.5 m resolution, while the PHOTO DEM over-estimated elevation by approximately 10 cm.

Despite customization and overall higher accuracies of PHOTO DEMs, they had more outliers than LiDAR DEMs, where the quantity of outliers was influenced by the topography at the study sites. Indeed, a major drawback of PHOTO is the influence of external factors on DEM accuracy, where bumpy terrain is known to cause problems due to the projection of shadows and lighting irregularities [23]; sites with more complex topography are likely to produce DEMs with more outliers and inaccuracies than sites with smoother terrain [27]. These errors and outliers can be edited out during post-processing, as was done here, but this does involve considerable time and good knowledge of DEM-processing systems, such that it might not be feasible for non-experts.

DEM extent affected the calculation of derived variables, such that we found between LiDAR and PHOTO DEM-derived variables. Though DEM extents of both technologies surpassed the targeted areas, LiDAR captured much larger extents than did PHOTO and included the surrounding topography (mountains, cliffs, etc.). Inclusion of surrounding topography is particularly important for variables such as total irradiance in June (Ti06), whose accuracy depends on the inclusion of the overall surface orientation and total incoming solar radiation, which may be affected by obstructing objects, such as mountains or boulders [14]. Limited extent for the PHOTO DEM also affected the number of times it could be generalized before border pixels were lost over the target site.

The relevance of DEM-derived variables from each technology depended on study site characteristics. At Para, SDM performance was improved with LiDAR-derived variables, indicating the importance of having a DEM with a large enough extent to include the site's surrounding mountainous topography. At Martinets, PHOTO DEM-derived variables at the mid-range resolutions were more relevant in predicting *A. alpina* distributions than LiDAR derived variables at matching resolutions. This may reflect the reduced influence of surrounding mountains at Martinets, as well as the bias for PHOTO variables due to potential snow cover in the LiDAR DEM.

### *4.3. Recommendations*

Both LiDAR and PHOTO are valid technologies for acquiring DEMs in alpine regions, particularly when collected with the aim of deriving ecologically relevant variables. The choice between which technology to use depends on the characteristics of the study site, the extent, and spatial resolution required, as well as budget and planned frequency of surveys required to be carried out. The key characteristics and technical properties of LiDAR and PHOTO technologies are summarized in Table 3. We acknowledge that though LiDAR is now available on drones and is thus cheaper than airplane-based LiDAR, the drones required to carry the LiDAR and its battery means that it remains expensive when compared with drone-based PHOTO (LiDAR on drones > US$10k; PHOTO on drones > US$5k), while raising the same issues as drone-based PHOTO of limited extents at arguably too fine resolutions (1–3 cm resolutions). As such, we focus only on comparing airborne LiDAR and drone-based PHOTO.

### 4.3.1. Characteristics of the Study Site

LiDAR is preferred for sites with moderate vegetation and complex terrain, as the laser is not impacted by illumination and can penetrate vegetation. As PHOTO produces digital surface models (DSMs), the site must have minimal vegetation and obstructions present. In homogenous terrain with minimal vegetation, PHOTO is just as good as LiDAR.

We recommend that the extent of DEMs cover larger areas than simply the target site, particularly when surrounding topography is likely to influence variables, and to ensure that information around the border is not lost when generalizing DEMs to coarser resolutions.

**Table 3.** Key differences between airborne LiDAR and drone photogrammetry (PHOTO) technologies to acquire DEMs, with regards to technical aspects and data characteristics. An additional column 'Pref.' indicates whether LiDAR or PHOTO are preferred for a given aspect, where "Both" is marked in cases where the preferred technology is context dependent. See [15,23,29,30,63] for review articles of these two technologies.

| | **LiDAR** | **PHOTO** | **Pref.** |
|---|---|---|---|
| **Data acquisition** | | | |
| Sensor | Active (laser and sensor) | Passive (images) | Both |
| Vehicle used | Fixed-wing vehicle or helicopter | Drones | Both |
| Flight details | Faster and longer flight, with 20–30% overlap, more complicated flight planning | Slower and shorter flight, with 60–90% overlap, more simple flight planning | Both |
| Area covered | Regional | Local | Both |
| Flight conditions | Light- and weather-independent | Light- and weather-dependent (diffused light to avoid shadows, dry weather, low winds) | LiDAR |
| Terrain type | Suited to most terrain types | Suited to open areas with smooth, visually distinct objects | LiDAR |
| Processing time | Fast/direct | Long/slow | LiDAR |
| Cost | Aircraft: ~US\$680–1400/km$^2$ (outsourced service) | Drone: >US\$5000 (for complete drone and sensor purchase—acquisition for own use) | PHOTO |
| Software | Open source available (e.g., PDAL); Software licenses start at ~US\$150/month (e.g., TerraScan) | Open source available (e.g., MicMac); Software licenses start at ~US\$200/month (e.g., Pix4D) | Both |
| **Data characteristics** | | | |
| DEM produced | DTM + DSM [1] | DSM (DTM if little or no vegetation) | LiDAR |
| Data presentation | Monochrome, points only; additional camera can be used for photos | Color and near-infrared images, photos | PHOTO |
| Land classification | Points classified based on reflection and return of laser | Pixels classified later based on point height | LiDAR |
| Data resolution | 50 cm depending on sensor and flight height | 1–3 cm depending on sensor and flight height | PHOTO |
| Feature preservation | May miss some geomorphological features | High performance in preserving geomorphological features | PHOTO |
| Derived variables | Produces more variables | Produces fewer variables due to reduced coverage of surrounding topography | LiDAR |
| **Data accuracy** | | | |
| Accuracy | Better vertical than horizontal | Better horizontal than vertical | Both |
| Characteristics | Accuracy may not be uniform over survey area | More homogeneous within the image format | Both |
| Control points | Low number for validation | High number for photo matching and validation | LiDAR |

[1] DTM = Digital terrain model; DSM = Digital surface model.

Given that SDM performance seems to reduce when input variables are at very high resolutions, in addition to the increase in computational time when using such detail, we recommend carefully considering the necessity and use of VHR variables finer than 0.5 m in ecological alpine studies of plants, and we strongly encourage implementing a multiscale approach to optimize the spatial resolution of derived variables.

### 4.3.2. Logistics

Acquiring DEMs using the PHOTO method requires more pre-flight planning, particularly as PHOTO is heavily influenced by weather and light conditions, as well as vegetation and other non-topographical obstructions. Light conditions are of particular note, as optimal PHOTO results are obtained with diffused light (e.g., on an overcast day or with the sun low on the horizon), as full sunlight results in high contrast and can lead to errors in the DEM. The effect of light is amplified in alpine regions where fewer atmospheric

particles means less light scattering and higher contrasts than at sea level, resulting in more errors. In regions of complex topography where vegetation, lighting, and weather may pose problems, LiDAR is likely to be the more accurate option. For more information, see reviews, such as [25,29,30].

Despite logistical difficulties in acquiring PHOTO data, it remains the cheapest option for producing DEMs—an advantage when an area needs frequent surveys. Where LiDAR DEMs can cost upwards of US$680–1400 per km$^2$ depending on whether it is specifically commissioned or already available, costs of purchasing one's own PHOTO equipment (drones and sensors) begin at around US$5000, such that cost per km$^2$ reduces with use. Post-processing is required for both LiDAR and PHOTO technologies, and while open-source software are available, professional software licenses that improve speed and quality start at around US$150–200 per month, or several thousand USD for a permanent license, for both technologies.

### 4.3.3. Environmental Variables

Here, we began with 23 commonly derived variables, and after assessing correlation using a Spearman's rank threshold of 0.8, we retained eight uncorrelated variables, prioritizing primary terrain attributes and variables that were deemed to be more ecologically meaningful to high-altitude alpine plants, such as soil pH, solar radiation, and wind exposure [48]. While we used an example of eight variables, there are a plethora of other variables available for use in ecological studies. As the variable selected can alter the outputs of ecological models, they must be carefully selected [64]. Indeed, [13] have compiled a table that groups frequently covarying variables together so that the reader can select six to seven terrain attributes that likely capture about 70% of surface structures across a site. We recommend that the reader consults this table prior to selecting variables for their own research.

When selecting variables for ecological models, one must first ensure independence between variables to avoid redundancy [13,19]. One option is to reduce the dimensionality of the data by performing a principal component analysis (PCA) with all variables that may be relevant to the study, then use the coordinates of the PCA-components as uncorrelated input predictor variables in the model. While this method inherently ensures independence between input variables, there is a loss of specific information about the environmental variables that might be driving patterns seen in the study. An alternative is to assess the collinearity of the derived variables, and to select only those that are uncorrelated at a certain threshold (here we used $|r_S| \geq 0.8$ as suggested by [48]). In using this latter option, however, it should be noted that collinearity between variables may vary with spatial resolution, so we recommend reassessing variable collinearity at the spatial resolutions intended to be used.

### 4.3.4. Evaluation

Prior to deriving variables, we recommend assessing the accuracy of DEMs as an indication of the reliability of the final model results. We recommend consulting [46], who have detailed outlines on best practices for evaluating DEMs. Derived variables can be evaluated by comparing them with field-collected environmental data from sensors or ecological species indicator values. In our case, a multiscale VHR-DEM was evaluated in this way in a close area during the pilot phase of the GENESCALE project [15].

## 5. Conclusions

Advancements in LiDAR and photogrammetry (PHOTO) technologies are opening new doors for ecologists to model alpine habitats at very high spatial resolutions (VHR). Reductions in costs and improvements in accuracy and ease-of-use, particularly for PHOTO, has allowed researchers to obtain VHR-digital elevation models (DEMs) at finer than 1 m spatial resolution, which are being used to produce VHR-derived environmental variables. Here, we demonstrate that PHOTO DEMs rivalled the accuracies of LiDAR up to a spatial

resolution of at least 1 m. Despite this, the reduced extent of the PHOTO DEMs had consequences on the calculations of derived variables, with subsequent effects on their relevance in species distribution models (SDM). Indeed, for the heterogeneous site situated in a narrow valley, PHOTO-derived variables resulted in reduced SDM performances, likely attributable to its reduced accuracy and extent, leading to inaccuracies in some places. At the homogenous site in a wider valley, however, SDM performance based on PHOTO-derived variables generally had higher predictive powers than those of LiDAR, as they sufficiently covered the relevant terrain, and because the LiDAR DEM was affected by snow cover.

We support the use of the cheaper PHOTO technology, as long as the researcher acknowledges this technology's drawbacks across complex terrain with obstructions and that certain weather conditions may cause issues for PHOTO sensors. However, and regardless of the technology used, we do not recommend using VHR-DEMs finer than 0.5 m resolution for alpine plant research. We encourage researchers to implement a multiscale framework to appropriately assess ecological relevance of derived variables, and we urge researchers to carefully select variables prior to obtaining DEMs to ensure sufficient coverage over the study site.

**Supplementary Materials:** The following supplementary tables and figures are available online at https://www.mdpi.com/article/10.3390/rs13081588/s1, Table S1: Description and parameters for Elevation and 23 digital elevation model (DEM)derived variables, Table S2: Student *t*-tests for MaxEnt analyses, Figure S1: Quantile-quantile (Q–Q) plots for digital elevation model (DEM) vertical error, Table S3: Summary results ranking MaxEnt species distribution models (SDM) to determine optimal parameters, Table S4: Summary statistics of digital elevation models (DEM) vertical error, Figure S2: Normalized median absolute deviation (NMAD) of digital elevation model (DEM) vertical error, Table S5: Spearman's correlation $r_s$ between pairs of independent derived variables, Figure S3: Scatterplots of derived variables from LiDAR and photogrammetry.

**Author Contributions:** Conceptualization, A.S.G., K.L., M.K. and S.J.; Data curation, A.S.G., K.L., A.R. and M.K.; Formal analysis, A.S.G., K.L., E.R. and S.J.; Funding acquisition, M.K., F.G., C.P. and S.J.; Methodology, A.S.G., K.L. and S.J.; Project administration, M.K., F.G., C.P. and S.J.; Resources, K.L., A.R. and M.K.; Supervision, S.J.; Visualization, A.S.G.; Writing—original draft, A.S.G.; Writing—review & editing, A.S.G., K.L., E.R. and S.J. All authors have read and agreed to the published version of the manuscript.

**Funding:** The GENESCALE project was funded by Schweizerischer Nationalfonds zur Förderung der Wissenschaftlichen Forschung, Grant/Award Number: CR32I3_149741.

**Data Availability Statement:** The data presented in this study are openly available at Zenodo at doi:10.5281/zenodo.4575251 [39].

**Acknowledgments:** The authors would like to acknowledge François Felber for his contribution to funding acquisition. We would also like to thank Nicolas Delley and Stéphane Cretegny for their work in collecting the photogrammetry DEMs in the field and undertaking the post-processing, as well as Lolita Ammann and Jessica Joaquim for their assistance in the field. The authors also thank Oliver Selmoni for comments and suggestions in data analysis, as well as Emmanuel Clédat, Kyriaki Mouzakidou and Gabriel Laupré for invaluable discussions on LiDAR and photogrammetry technologies. We thank the anonymous reviewers for their constructive comments that improved the quality of this work.

**Conflicts of Interest:** The authors declare no conflict of interest.

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
