# Peer review of "Multiscale Very High Resolution Topographic Models in Alpine Ecology: Pros and Cons of Airborne LiDAR and Drone-Based Stereo-Photogrammetry Technologies"

_remotesensing, doi:10.3390/rs13081588_

Round 1

Reviewer 1 Report

Overall Impression
This manuscript is easily one of the best-written manuscripts I’ve reviewed in a long time. The abstract and introduction clearly state the study’s objectives, and the methods and results are depicted with equal clarity, while the discussion firmly contextualizes the study’s findings in the literature. The authors’ recommendations are clear while retaining nuance (such as a caveat to reconsider the multicollinearity of environmental variables when choosing to rescale resolutions of the supporting DEMs). I congratulate the authors on their study and its description, and I highly recommend the manuscript for publication. Minor suggestions for improvement of the figures/tables and some general proofreading follow.

General Comments
I find it odd in Figure 1 and 2 that the westernmost study area is placed to the right of the easternmost one. Perhaps change this order?

In Table 3, it would be good to add a “both” or “tie” label in the final column for the blank cells, as I’d originally assumed a typographical error was the cause.

Line by Line Comments
260: “commonylyused” needs to be separated.
475: “finer that” should be “finer than”

Author Response

Overall impression

This manuscript is easily one of the best-written manuscripts I’ve reviewed in a long time. The abstract and introduction clearly state the study’s objectives, and the methods and results are depicted with equal clarity, while the discussion firmly contextualizes the study’s findings in the literature. The authors’ recommendations are clear while retaining nuance (such as a caveat to reconsider the multicollinearity of environmental variables when choosing to rescale resolutions of the supporting DEMs). I congratulate the authors on their study and its description, and I highly recommend the manuscript for publication. Minor suggestions for improvement of the figures/tables and some general proofreading follow.

We thank the reviewer for their very kind and insightful comments. Please find responses to specific modifications below.

General Comments

I find it odd in Figure 1 and 2 that the westernmost study area is placed to the right of the easternmost one. Perhaps change this order?

We agree that this suggestion improves clarity, and as such we have modified both Figures 1 and 2 so that the eastern-most site is on the right-hand side for both. We have also modified Table 1 to maintain consistency with these figures and updated all legends accordingly.

In Table 3, it would be good to add a “both” or “tie” label in the final column for the blank cells, as I’d originally assumed a typographical error was the cause.

Thank you for this feedback. We have now included more information in the ‘Preferred’ column of Table 3 to avoid ambiguity, and we have included an additional sentence in the table caption for clarity.

Line by Line Comments

260: “commonylyused” needs to be separated.

Done.

475: “finer that” should be “finer than”

Done.

Reviewer 2 Report

Dear Authors,

This is interesting and very practical LiDAR and PHOTO topographic model comparison in mountain terrain. The work is very systematic, detailed, tidy and well done. It gives knowledge and hints for the reader. I appreciate the quality of the work. Another advantage of the work is that you share your data. Congratulations. My decision is: accept in the present state.

Sincerely yours,

Reviewer

Author Response

Dear Authors,

This is interesting and very practical LiDAR and PHOTO topographic model comparison in mountain terrain. The work is very systematic, detailed, tidy and well done. It gives knowledge and hints for the reader. I appreciate the quality of the work. Another advantage of the work is that you share your data. Congratulations. My decision is: accept in the present state.

Sincerely yours,

Reviewer

We thank the reviewer for their very kind and supportive comments.

Reviewer 3 Report

This paper compares DEM-derived variables produced by airborne LiDAR data and UAV stereo images. DEMs were obtained at spatial resolutions from 6.25cm to 8m for images and from 50cm to 32m for LiDAR for the arctic-alpine plant in two valleys in the western Swiss Alps. The authors claimed that “PHOTO DEMs rivalled the accuracy of LiDAR, putting the current paradigm of LiDAR being the more accurate of the two methods into question.”

There are two main issues in this article:

  1. Line 194-195: how did the authors use points as control points (24 and 13), and then used them again in accuracy assessment? I think it causes bias in the results; I would like to see the comparison results excluding the control points.
  2. The authors already mentioned the possible logical reasons in Line 408-419 of the claim in Line 25-27. I think it is not acceptable to leave as it is in the abstract. The authors should mention the reasons in the abstract, or modify the sentence in the abstract.

Comments:

  1. Table1: add units of slope, East, …
  2. Line 171: what is the type of neighborhood interpolation did you use?
  3. Line 172: how did you define the finest resolution of DEM for both methods?
  4. Line 184-186: justify the multiple scales used, why these values? for both methods.
  5. Table S4 is unreadable, please separate both sites
  6. Support conclusions with results

Author Response

This paper compares DEM-derived variables produced by airborne LiDAR data and UAV stereo images. DEMs were obtained at spatial resolutions from 6.25cm to 8m for images and from 50cm to 32m for LiDAR for the arctic-alpine plant in two valleys in the western Swiss Alps. The authors claimed that “PHOTO DEMs rivalled the accuracy of LiDAR, putting the current paradigm of LiDAR being the more accurate of the two methods into question.”

We thank the reviewer for their insightful comments that have helped to improve the clarity and integrity of our manuscript. Please find our responses to specific comments below.

There are two main issues in this article:

  1. Line 194-195: how did the authors use points as control points (24 and 13), and then used them again in accuracy assessment? I think it causes bias in the results; I would like to see the comparison results excluding the control points.

The reviewer raises an excellent point, and we apologise for this oversight. Indeed, we now see and acknowledge that some bias towards photogrammetry DEM accuracy was introduced into our results by including the ground control points (GCPs) into assessments, as these GCPs were initially collected and used to process the photogrammetry DEMs.

To rectify this, we have re-assessed the accuracies of all DEMs using only the geo-referenced points of the loggers and plants, and we have updated Figure 3, Figure S1, Figure S2, and Table S4 accordingly, as well as their captions. After removing GCPs from the assessments, there was little change in DEM accuracies at Para. However, at Martinets, photogrammetry DEM accuracies reduced slightly, such that at an additional outlier was detected at the finer resolutions, and variation of the photogrammetry DEM errors increased at resolutions coarser than 1m.  We have made the necessary adjustments to the text in Section 3.1 (lines 286-306) in the results and Section 4.2 in the Discussion (lines 409-420) to reflect the updated results.

  1. The authors already mentioned the possible logical reasons in Line 408-419 of the claim in Line 25-27. I think it is not acceptable to leave as it is in the abstract. The authors should mention the reasons in the abstract, or modify the sentence in the abstract.

We thank the reviewer for this comment, and we agree that the sentence at line 25-27 in the abstract lacked context and required modification, particularly in light of the new analyses. We have therefore modified this sentence (lines 25-27) to better reflect the updated results and to provide more appropriate context to our conclusions.

Comments:

1. Table1: add units of slope, East, …

We thank the reviewer for picking up this oversight. We have included units for Slope, Eastness and Northness, and noted in the footnote that VRM has no unit.

2. Line 171: what is the type of neighborhood interpolation did you use?

We have now clarified that we used linear interpolation with values from the nearest neighbouring cells (lines 175-177).

3. Line 172: how did you define the finest resolution of DEM for both methods?

We now explicitly state why we selected 0.5m for LiDAR DEMs and 6.25cm for photogrammetry DEMs (lines 177-180).

4. Line 184-186: justify the multiple scales used, why these values? for both methods.

We agree that more information was required, and we have modified Section 2.2.4 to include additional information to justify our choice of spatial resolutions (lines 191-197).

5. Table S4 is unreadable, please separate both sites

We thank the reviewer for this remark, and we agree that Table S4 was difficult to read. We have made changes to the formatting of the table and reduced information that is not referred to in the main text. We have also made minor modifications to the formatting of Table S2 and Table S5, so that results from photogrammetry DEMs come before those of LiDAR DEMs to ease interpretation.

6. Support conclusions with results

We agree that we did not include quantitative results in the conclusion. We have now modified Section 5 to include specific results and to highlight important interpretations for the reader (lines 538-547).

Reviewer 4 Report

Summary:The manuscript is about the pros and cons of airborne LiDAR and drone-based stereo-photogrammetry technologies. It has high application value, but the manuscript needs improvements to enhance. I have found some issues with the method proposed by this paper.

  1. Papers should be as accurate, true and objective as possible to describe the research results, in order to ensure the objectivity of the paper, the abstract and full text should be in the third person as far as possible. For example:

(1) line 268 - 270: “We note while that there may have … as per [60]” is not objective and accurate, the conclusion should be drawn by analyzing the data, not by “we note”.

(2) line 289-291: “We found that …”, how does the conclusion be found out, there needs to be data support and analysis process.

(3) line 309: “We showed …”, using “XX data show that …” would be more objective.

(4) line 367: “We found…”

  1. line 63: Whether it is possible to briefly explain the reasons for the selection of these environmental factors, or explain why these factors can be successfully used for related research.
  2. line 81: Can the author provide a specific value of the resolution here, not just multiples
  3. In section 2.1 of the article, can the author explain the reasons or methods for the distribution of recording points in number?
  4. line 275: “However, accuracy remained constant from 6.25cm through to 50cm.” is not accurate, from Figure 3, the accuracy varies a little from 6.25cm to 50cm.
  5. line 277: The figure on the right of Figure 3 is missing the axis label
  6. For Figure 3 to appear on one page, change the page. Or move the position of the caption (of Figure 3).
  7. It is expected that clearer figures 3 and 4 will be provided in this article.
  8. line 426:The author should explain the feasibility of dealing with errors by manual editing.

10.line 499 to line 501: It is best to briefly introduce the process or criteria for filtering these variables.

Author Response

Summary:The manuscript is about the pros and cons of airborne LiDAR and drone-based stereo-photogrammetry technologies. It has high application value, but the manuscript needs improvements to enhance. I have found some issues with the method proposed by this paper.

We thank the reviewer for their insightful comments that have helped to improve the clarity and integrity of our manuscript. Please find our responses to specific comments below.

  1. Papers should be as accurate, true and objective as possible to describe the research results, in order to ensure the objectivity of the paper, the abstract and full text should be in the third person as far as possible. For example:

(1) line 268 - 270: “We note while that there may have … as per [60]” is not objective and accurate, the conclusion should be drawn by analyzing the data, not by “we note”.

(2) line 289-291: “We found that …”, how does the conclusion be found out, there needs to be data support and analysis process.

(3) line 309: “We showed …”, using “XX data show that …” would be more objective.

(4) line 367: “We found…”

We thank the reviewer for this comment. As suggested, we have reworked the above-mentioned phrases (lines 274-275, 293-294, 313, and 369, respectively), as well as sentences throughout the abstract and main text (lines 32, 356, 397, 399, 358), to improve objectivity of the paper. We have not, however, rephrased all sentences to be in the third person, as we believe that in those cases the current phrasing aids in readability without compromising objectivity of the results, as is common in scientific articles.

  1. line 63: Whether it is possible to briefly explain the reasons for the selection of these environmental factors, or explain why these factors can be successfully used for related research.

We have rephrased the sentence on lines 64-68 to add clarity that these are examples of factors that can be used to represent actual environmental variables that might be important in ecological modelling.

  1. line 81: Can the author provide a specific value of the resolution here, not just multiples

We have now provided specific values for typical resolutions obtainable via LiDAR and photogrammetry to accompany the phrase ‘up to 10 times’ (lines 80-84)

  1. In section 2.1 of the article, can the author explain the reasons or methods for the distribution of recording points in number?

Thank you for this comment. We have now specified that the ground control points were selected as recognisable locations to produce photogrammetry DEMs (line 131), and that the plant and logger locations were selected within the context of the landscape genomics project to represent contrasting microhabitats across the sites (lines 136-137).

  1. line 275: “However, accuracy remained constant from 6.25cm through to 50cm.” is not accurate, from Figure 3, the accuracy varies a little from 6.25cm to 50cm.

Thank you for picking up on this phrasing. We have added ‘relatively constant’ into the sentence to better reflect the results (line 280).

  1. line 277: The figure on the right of Figure 3 is missing the axis label

We thank the reviewer for picking up this oversight on our part, and we have included axis labels for the Q-Q plots of Figure 3.

  1. For Figure 3 to appear on one page, change the page. Or move the position of the caption (of Figure 3).

We have modified the formatting of the article so that the captions align appropriately with the figures and tables throughout the main text.

  1. It is expected that clearer figures 3 and 4 will be provided in this article.

Thank you for picking up on this oversight. All figures in the main text are now provided at higher qualities as specified by the Journal (>300 dpi).

  1. line 426:The author should explain the feasibility of dealing with errors by manual editing.

We have adjusted the sentence so that it better reflects that performing these edits might be a challenge for non-experts (line 427-429).

  1. line 499 to line 501: It is best to briefly introduce the process or criteria for filtering these variables.

We thank the reviewer for this comment. We have added details on the selection of the eight uncorrelated variables as requested (line 502-506).

Round 2

Reviewer 3 Report

All comments have been addressed, but still for conclusions section, add results values.